

# Transgenerational effects enhance specific immune response in a wild passerine

Juli Broggi[1,2], Ramon C. Soriguer[3,4] and Jordi Figuerola[1,4]

[1] Wetland Ecology, Esatción Biológica de Doñana, CSIC, Sevilla, Spain
[2] Research Unit of Biodiversity (UMIB, UO/CISC/PA), University of Oviedo, Mieres, Spain
[3] Etologia y Conservacion de la Biodiversidad, Estación Biológica de Doñana, CSIC, Sevilla, Spain
[4] CIBER Epidemiología y Salud Pública (CIBERESP), Spain

## ABSTRACT

Vertebrate mothers transfer diverse compounds to developing embryos that can affect their development and final phenotype (i.e., maternal effects). However, the way such effects modulate offspring phenotype, in particular their immunity, remains unclear. To test the impact of maternal effects on offspring development, we treated wild breeding house sparrows (*Passer domesticus*) in Sevilla, SE Spain with Newcastle disease virus (NDV) vaccine. Female parents were vaccinated when caring for first broods, eliciting a specific immune response to NDV. The immune response to the same vaccine, and to the PHA inflammatory test were measured in 11-day-old chicks from their following brood. Vaccinated chicks from vaccinated mothers developed a stronger specific response that was related to maternal NDV antibody concentration while rearing their chicks. The chicks' carotenoid concentration and total antioxidant capacity in blood were negatively related to NDV antibody concentration, whereas no relation with PHA response was found. Specific NDV antibodies could not be detected in 11-day-old control chicks from vaccinated mothers, implying that maternally transmitted antibodies are not directly involved but may promote offspring specific immunity through a priming effect, while other immunity components remain unaffected. Maternally transmitted antibodies in the house sparrow are short-lived, depend on maternal circulation levels and enhance pre-fledging chick specific immunity when exposed to the same pathogens as the mothers.

## INTRODUCTION

Vertebrate mothers may provide a favourable growing environment and care, but additionally can transmit diverse components such as hormones (*Groothuis & Schwabl, 2008*), antioxidants (e.g., *Royle, Surai & Hartley, 2003*) or immunoglobulins (*Grindstaff, Brodie & Ketterson, 2003*; *Boulinier & Staszewski, 2008*; *Hasselquist & Nilsson, 2009*) that can have important phenotypic consequences on the developing embryo (*Mousseau et al., 2009*). These maternal effects constitute a major source of transgenerational phenotypic plasticity that can vary according to environmental heterogeneity (e.g., *Morosinotto et al., 2013*) and maternal condition (*Boulinier & Staszewski, 2008*). However, the way these transferred components interact with each other, and modulate the development of the offspring phenotype remains unclear (*Mousseau et al., 2009*).

Corresponding author
Juli Broggi, julibroggi@gmail.com

The vertebrate humoral immune response targets specific infectious agents by means of antibodies (Ab), and persists as immune memory allowing a faster response upon re-exposure to the same antigen. While adult birds can develop a humoral response in a few days (*Iseri & Klasing, 2013*), newborn individuals require some time to fully develop such capabilities and may rely on maternally transferred Ab while they develop their own endogenous response (*Hasselquist, Tobler & Nilsson, 2012*). Altricial avian small species are particularly vulnerable at hatching as they need to develop fast enough to be able to fledge in a few days time, and thus could be expected to rely preferentially on maternally transmitted Ab to develop their specific immunity. However, studies on the influence of maternal Ab on host development and fitness are scarce (*Boulinier & Staszewski, 2008*), despite the important influence individual immune condition exerts on pathogen susceptibility in wild organism populations (e.g., *Hochachka & Dhondt, 2000*). The influence of maternal Ab on the fitness and disease epidemiology of neonates in humans and domestic mammals is well documented, providing the offspring with transient immunity against the microbial infections that the mother has encountered (*Hasselquist & Nilsson, 2009*). In avian species, yolk-transmitted maternal Ab detectability ranges from days to few months after hatching, depending on the species (*Garnier et al., 2012*) and the initial amount transferred (*Grindstaff, 2010*). However, whether maternally transmitted Ab are enhancing the chick's subsequent response by a priming mechanism (e.g., *Grindstaff et al., 2006*; *Reid et al., 2006*), or blocking the chick's endogenous response (e.g., *Staszewski et al., 2007*; *Elazab et al., 2009*; *Staszewski & Siitari, 2010*) remains poorly understood.

Individuals exhibit substantial differences in their immune responses and rarely develop it maximally, which suggests there are some costs associated that individuals manage differently (*Viney, Riley & Buchanan, 2005*). Such individual variation in immune investment may arise from a variety of factors, not only adaptive adjustments but also constraints resulting from allocation conflicts with other physiological and life-history traits (*Ardia, Parmentier & Vogel, 2011*). Furthermore, individuals may invest differentially in various components of the immune system (*Lazzaro & Little, 2009*), depending on the pathogen identity (*Adamo, 2004*) or constraining nutritional or energetic factors (*Hõrak et al., 2006*), or alternatively these could vary in concert (*Ardia, 2005*; *Ahmed et al., 2007*), but see (*Forsman et al., 2008*).

Oxidative balance and carotenoids have been acknowledged among the factors mediating variation in the immune responses (*Hasselquist & Nilsson, 2012*). The intensity of the immune response is generally coupled with a shift in the oxidative balance and a decrease in carotenoid concentration that are considered detrimental for the individual (*Ardia, Parmentier & Vogel, 2011*). Oxidative stress is generated as a metabolic by-product resulting in damage to cell macromolecules (*Dowling & Simmons, 2009*). Organisms balance their oxidative stress by acquiring and producing antioxidants, and although most antioxidative activity is enzymatic, non-enzymatic antioxidants also play a relevant role in maintaining oxidative balance, particularly in blood (*Cohen & McGraw, 2009*). Immunity costs in terms of oxidative balance have been argued to underlie trade-offs between immunity and life-history traits, either by directly increasing oxidative stress (*Costantini & Møller, 2009*)

or by competing for antioxidants (*Monaghan, Metcalfe & Torres, 2009*), but see *Speakman & Garratt (2014)*.

Carotenoids are a diverse group of lipophilic molecules important for immunity and individual fitness, and since they cannot be synthesized *de-novo* by animals, necessarily need to be ingested or acquired during embryogenesis through maternal transfer (*Pérez-Rodríguez, 2009*). Avian mothers transmit carotenoids and antioxidants through the egg yolk, and after hatching through diet (*Blount et al., 2003*; *McGraw & Ardia, 2003*). These components interact synergistically (*Bédécarrats & Leeson, 2006*; *Koutsos, García López & Klasing, 2007*) and likely stimulate the development of the offspring own immune phenotype (*Simons, Cohen & Verhulst, 2012*). However, little is known on how such maternal effects may interact with each other, especially on wild and non-model species (*Hasselquist & Nilsson, 2012*).

In this study we explored whether maternal effects modulate offspring specific immune response in a wild breeding house sparrow (*Passer domesticus*, Linnaeus 1758) population. Using a vaccine to elicit an immune response to a viral antigen, we analysed how maternal exposure during first brood affects transmission of antibodies to the following brood, their offspring development and immune response when exposed to the same antigen during the critical post-hatching period before fledging. We additionally measured chick PHA-induced inflammatory response, to find out whether mother immune condition affected their offspring specific humoral response, or affected more aspects of offspring immunity. Furthermore, we studied the effect of other maternally-transmitted components (antioxidative and carotenoids) which are likely to affect the development of the offspring immune phenotype. We predict vaccinated mothers to positively affect their offspring specific immunity, and this effect to be enhanced by higher levels of maternal antioxidants, antioxidative capacity and carotenoids in blood.

## MATERIALS AND METHODS

### Study area and model species

The house sparrow is a small-sized (20gr) granivorous passerine that breeds in close association with humans. House sparrows are commonly exposed to a variety of pathogens in the wild, including several epizootic pathogens resulting from the close association with humans and livestock e.g., Newcastle disease virus or *Salmonella* (*Anderson, 2006*). The Newcastle disease virus (NDV) is a worldwide distributed avian paramyxovirus that causes a highly contagious disease, representing a severe problem for the poultry industry and also wild fauna (*Alexander, 2009*). The virus is circulating in the study area as NDV antibodies were detected previously in 11 out of 81 individuals analyzed (*Broggi et al., 2013*).

The study area is located in a private land surrounded by farmland and mixed forest, la Cañada de los Pájaros (37°14′N, 6°07′W) in Sevilla, SE Spain. The study population is about 100 pairs breeding naturally in wooden nestboxes at an average height of 2 m and within an area of 10 ha. Females lay up to 4 broods per year, usually in the same nestbox, with an average clutch of 4.5 eggs that hatch asynchronously. Chicks fledge at the age of 12–15 days if undisturbed. Breeding season starts in early April and lasts until the end of August (J Broggi, 2010, unpublished data).

## Experimental approach

From April 1st (the beginning of the breeding season) until August 2010, nestboxes were checked every second day to record breeding parameters. Breeding females were captured at the nest when chicks were older than 8 days to prevent nest desertion. Newly captured females were randomly assigned to the treatment (subcutaneous injection of 0.2 ml of a commercial inactivated NDV vaccine HIPRAVIAR® BPL2) or a control group (injection with 0.2 ml of PBS), following the results of a pilot study in the same population (*Broggi et al., 2013*). Before treatment, blood was sampled from the jugular vein (0.2 ml) and kept cool (~4 °C) for less than 12 h before centrifugation (20 min at 4,000 rpm). Cellular phase and sera were stored separately at −20 °C for later analyses (see below), and wing, tarsi and body mass were measured. Females were allowed to complete the first breeding attempt without further manipulation to minimise disturbance. Recaptured females were treated as in their first capture. During the next breeding attempt the chicks were weighed (to 0.1 g) on their 4th day of age, and were inoculated subcutaneously with either NDV vaccine (0.1 ml) or a control treatment (PBS). Chicks within each brood were ordered by body mass and inoculated alternatively with vaccine or control treatment, switching the starting treatment in each different brood. Chicks were recognized by innocuous paint in their claw, and were repainted until they were marked with aluminium rings when ~6 days of age. Due to hatching asynchrony, chick age differed within broods in up to 4 days (J Broggi, pers. obs., 2010). When average brood age was 11 days, chicks were weighed, and their tarsi and wing length measured (to 0.1 mm). Blood samples were taken from the chick's jugular vein (0.1 ml) and processed as with adult female samples. Finally, chicks were subjected to a phytohaemagglutinin (hereafter PHA) immune challenge before being released in their nestbox. On the following day, chicks were re-measured (see below for details on the PHA immune challenge). Sex of the chicks was determined by molecular techniques based on DNA obtained from blood samples (*Fridolfsson & Ellegren, 1999*). Females were recaptured on the second breeding attempt, and blood was sampled to measure blood metabolites and NDV-Ab concentration (see below). Each experimental female was included once in the dataset, and whenever captured in more than two consecutive nesting events (two cases), only the last breeding episode was included in order to ensure a high female response to vaccination at the time of egg-laying. On average, experimental females were challenged 3–10 weeks before egg-laying of the following clutch, well-within the antibody circulation peak after vaccination (*Broggi et al., 2013*; *Midamegbe et al., 2013*). Altogether, the dataset consisted of 88 chicks from 25 different females (13 control vs. 12 vaccinated). Control chick sample size consisted of 21 individuals originating from control mothers and 19 from vaccinated mothers. Vaccinated chick sample size consisted of 25 individuals originating from control mothers and 23 originating from vaccinated mothers (Table 1).

## Immunological measurements

We used haemagglutination inhibition test (HI) to assess NDV-Ab concentration in sera. The test sera (25 µl) were sequentially diluted in PBS from 1/2 to 1/640 and 4HA units of antigen HIPRAVIAR®-CLON E.Newcastle, clon CL/79 were added to each dilution. The mixture was added to 50 µl of chicken RBC's and after 30 min at room temperature

**Table 1  Body mass and blood metabolite profiles for the different experimental treatments with their corresponding sample size.** Body mass increase of house sparrow chicks from vaccination at four days of age to blood sampling at ten days of age, and body mass of house sparrow mothers according to the different treatments (C = Control, V = Vaccinated). Change is mass is expressed as the least squared means from a GLM with treatments and the number of days elapsed as a covariate. Different treatments correspond to the combination of mother (Mo) and chicks' (Ch) treatment: control chicks from control mothers (MoC ChC); control chicks from vaccinated mothers (MoV ChC); vaccinated chicks from control mothers (MoC ChV); and vaccinated chicks from vaccinated mothers (MoV ChV). Mean values with the corresponding standard error (SE) are provided for the different blood metabolite parameters (Total antioxidant capacity (TAC); Carotenoids (CAR); Total protein (TPR); Uric Acid (UAC)), for chicks and mothers on different treatments. Sample sizes are given within parentheses.

| Chicks | Mass change ± SE (g) | TAC ± SE (μmol/L) | CAR ± SE (mg/L) | TPR ± SE (mg/dL) | UAC ± SE (mg/dL) |
|---|---|---|---|---|---|
| MoC ChC | 6.98 ± 0.70 (21) | 931.34 ± 111.05 (21) | 15.62 ± 2.26 (16) | 2.23 ± 0.16 (21) | 7.03 ± 0.62 (21) |
| MoC ChV | 7.14 ± 0.63 (25) | 721.07 ± 103.88 (24) | 14.70 ± 2.02 (20) | 2.14 ± 0.15 (24) | 6.82 ± 0.58 (24) |
| MoV ChC | 5.52 ± 0.73 (19) | 890.97 ± 116.75 (19) | 10.38 ± 2.26 (16) | 2.10 ± 0.17 (19) | 6.81 ± 0.65 (19) |
| MoV ChV | 4.99 ± 0.67 (23) | 888.00 ± 111.05 (21) | 12.19 ± 2.08 (19) | 2.36 ± 0.17 (21) | 6.58 ± 0.62 (21) |
| **Mothers** | **Mass ± SE (g)** | | | | |
| Mo C | 26.25 ± 0.46 (13) | 1101.48 ± 139.12 (12) | 8.46 ± 3.19 (12) | 3.09 ± 0.22 (12) | 14.15 ± 1.78 (12) |
| Mo V | 26.11 ± 0.48 (12) | 963.58 ± 145.31 (11) | 10.16 ± 2.88 (12) | 3.02 ± 0.23 (11) | 14.84 ± 1.85 (11) |

checked for agglutination. NDV-Ab concentration was scored as the highest dilution where agglutination was observed. We used commercial positive and negative controls (VLDIA053 HAR-NDL, NDV strain La Sota and VLDIA030 SPF-CH-Chicken negative), further details can be found in *Broggi et al. (2013)*. None of the females at first capture nor control females at second capture presented NDV titers higher than 1/8, the lower limit to consider a serum positive to NDV according to standard laboratory practices (*OIE, 2015*).

At 11 days of age, chicks were challenged with PHA in the patagia. PHA is a mitogen of vegetal origin that when injected intradermally induces an immune response mainly reflecting individual pro-inflammatory potential (*Vinkler, Bainova & Albrecht, 2010*). Birds were injected in the right wing web with 50 μl of 5:2 PHA-P (L-8754 Sigma-Aldrich) in PBS following *Smits, Bortolotti & Tella (1999)*. Patagium width was measured at the point of injection (to the nearest 0.01 mm) just prior to and at least 24 h from challenge, using a pressure sensitive micrometer (Baxlo Precision S.L.), and the difference was used as PHA response thereafter. Time between measurements averaged 40.20 ± 0.81 h, range = 25–45 after challenge. PHA-challenge elicits an inflammatory response that reaches its maximum after ∼6 h from injection, and can last up to 72 h (*Navarro et al., 2003*). Three measures were taken consecutively, removing the micrometer each time, and the average was used in the analyses (intra-class correlation $R > 0.98$).

## Measurement of blood metabolites

Carotenoid concentration in sera was measured by means of N-1000 NanoDrop spectrophotometer at 450 nm, at the maximum reflectance point for lutein, which is the main circulating carotenoid found in passerines (*Pérez-Rodríguez, 2009*). Total antioxidant capacity (TAC) in sera was measured as described in (*Erel, 2004*), implemented in Cobas INTEGRA Chemistry autoanalyser. Recent studies point out that TAC is mostly representative of the water soluble components of the antioxidative balance, and in

combination with other fat-soluble antioxidants provides a more complete image of the antioxidant system (*Cohen & McGraw, 2009*). Uric acid and total protein were measured according to standard methods implemented on a Cobas Integra 400 plus autoanalyzer with Roche reagents. Uric acid is a common circulating antioxidant generated as a by-product of metabolism that accounts for an important portion of the antioxidant capacity in blood, whereas total protein in sera is a standard diagnostic measure of nutritional condition and may also reflect serum antioxidative activity (*Roche et al., 2008*). Repeatability for carotenoids, Uric acid, Total protein and TAC in sera, as assessed by the intraclass correlation of repeated blind measures of the same samples was higher than 0.97 for all parameters.

## Statistical methods

NDV-Ab concentration (expressed as the base 2 logarithm of the inverted dilution factor) from 11-day-old chicks was analyzed as dependent variable in a generalized linear mixed model (GLMM), with normally distributed error and identity link. Mother and chick's treatments were included as main effects and nestbox as a random factor in all models. Mother's final NDV-Ab concentration (when caring the second brood), and time between treatment and sampling were considered as covariates in both chick ($7.72 \pm 0.19$ days, range 5–12) and female ($37.64 \pm 0.72$ days, range = 24–65) measurements, as differences arose due to hatching asynchrony and capture success respectively. Likewise, in addition to main effects and the random factor, time between measurements was included as covariate in the models for PHA response and mass change from challenge to sampling, and otherwise models were identical. Chick survival from 4th day of age until fledging was analysed with a GLMM with binomially distributed error and logit link. Chick and mother treatments were included as fixed factors, together with body mass at the time of chick's treatment as covariate and nest as a random factor.

Afterwards, several parameters were tested in the previous models as covariates i.e., breeding parameters (hatching date and clutch size), sex, biometric data (body mass and tarsus length) and blood metabolites (TAC, Uric acid, Total protein and carotenoids). Different models were built on NDV-Ab concentration in chicks and mothers; chick PHA response, mass change and chick survival as dependent variables. Only breeding parameters were included as covariates in the models on chick survival as no sex, biometric nor blood metabolite information were available from non-surviving chicks. Due to limited sample size, covariates and their interactions with the main effects were included sequentially, and retained whenever significant or as judged by AIC or Generalized Chi$^2$ to avoid model over-parametrization. Initial and final models are presented with the corresponding fit statistics and variable parameters (Estimate, F, DF and P values), and non significant predictors are shown with their corresponding values when removed from the final model (Supplemental Information 1).

Least squared means are provided for categorical predictors and slope estimates for covariates ± standard error. Sample size varied among tests due to differential survival and available sera for laboratory analyses (Table 1). Degrees of freedom for fixed effects were adjusted by the between-within approximation that accounts for within-subject changes of

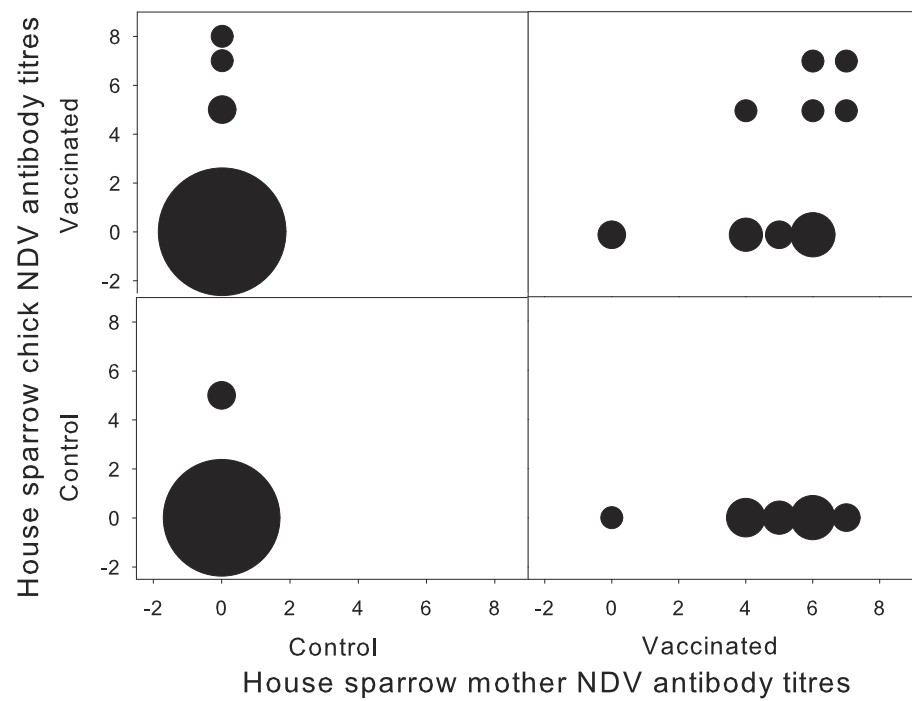

**Figure 1 Concentration of NDV antibodies in house sparrow with respect to their mother, for each experimental treatment.** Newcastle disease virus (NDV) antibody titres for house sparrow chicks and their mothers, in relation to the different experimental treatments ($N = 1$–20). Sizes of the circles correspond to sample size. Antibody titres are expressed as the log of the inverse of the dilution factor.

any fixed effect and divides the residual degrees accordingly (*Schluchter & Elashoff, 1990*). Blood metabolite variables were normalized by means of $\log(x + 1)$ transformation, and residuals from all models on NDV-Ab concentration, PHA response and mass change followed normality. All analyses were performed with procedure GLIMMIX SAS 9.2. (SAS Institute Inc. 2009). Raw data is available at http://hdl.handle.net/10261/127825.

## RESULTS

Female NDV-Ab concentration before challenge were similar between treatments ($F_{1,14} = 0.86$; $P = 0.37$), and no differences between experimental and control groups were found before challenge for any of the physiological (Table 1) or reproductive parameters studied. As expected, female NDV-Ab concentration increased significantly in vaccinated individuals with respect to control ones (Control: $0.00 \pm 0.38$ vs. Vaccinated: $4.70 \pm 0.41$; $F_{1,20} = 70.67$; $P < 0.0001$) (Fig. 1). Neither pre-challenge NDV-Ab concentration ($F_{1,12} = 0.39$; $P = 0.54$), days between vaccination and re-sampling ($F_{1,13} = 0.07$; $P = 0.80$), the physiological nor the reproductive parameters considered (all $P > 0.5$) were related to post-challenge NDV-Ab concentration.

Chick NDV-Ab concentration at 11 days of age was independent of maternal treatment ($F_{1,17} = 0.06$; $P = 0.81$). Surprisingly, chick NDV-Ab concentration was independent from chick treatment ($F_{1,19} = 3.83$; $P = 0.07$), and the interaction with maternal treatment ($F_{1,19} = 0.52$; $P = 0.48$). Time between chick vaccination and sampling was not significant

($F_{1,17} = 0.26$; $P = 0.615$). However, when considering female NDV-Ab concentration as a covariate, the interaction between female NDV concentration and chick's treatment appeared significant ($F_{1,53} = 6.61$; $P = 0.01$), together with days between chick and mother sampling (Estimate $0.324 \pm 0.121$; $F_{1,17} = 7.20$; $P = 0.02$). NDV-Ab concentration of vaccinated chicks tended to increase with female NDV-Ab concentration, whereas this relation was not significant in control chicks (Control: $-0.11 \pm 0.14$; $t_{18} = -0.81$; $P = 0.43$ vs. Vaccinated: $0.23 \pm 0.13$; $t_{18} = 1.75$; $P = 0.09$). Neither sex, biometric measurements nor the breeding parameters were associated with NDV-Ab concentration in chicks (all $P > 0.2$). Vaccinated chicks from vaccinated mothers developed higher specific humoral response, but only when originating from mothers exhibiting high NDV-Ab titres (Fig. 1).

On the other hand, when including blood metabolites as covariates on the previous model, only carotenoids and TAC were related to chick NDV-Ab. Carotenoids decreased with increasing NDV-Ab concentration in chicks, independently of the treatment (slope: $-1.35 \pm 0.47$; $F_{1,39} = 8.11$; $P < 0.01$). However, the relation between NDV-Ab and TAC changed slightly according to the chick treatment, the interaction being significant ($F_{1,50} = 5.95$; $P = 0.02$). NDV-Ab concentration of vaccinated chicks decreased with TAC, whereas the relation was not significant in control chicks (Control: $-0.34 \pm 0.59$; $t_{50} = -0.59$; $P = 0.56$ vs. Vaccinated: $-2.08 \pm 0.45$; $t_{50} = -4.66$; $P < 0.01$). Chick's concentration of NDV-Ab was negatively related to carotenoids in blood, and in the case of vaccinated chicks also to TAC in blood (Fig. 2 in Supplemental Information 1). Uric acid or total protein were unrelated to chick's NDV-Ab concentration (all $P > 0.1$).

PHA in chicks was unrelated to maternal ($F_{1,9} = 0.01$; $P = 0.93$) or chick ($F_{1,21} = 0.02$; $P = 0.88$) treatments, even when accounting for chick body mass or change in body mass between challenge and measurement (hereafter body mass change), and time between such measurements (all $P > 0.2$). Furthermore, none of the breeding parameters, biometric measurements nor blood metabolites considered were significantly related to PHA response (all $P > 0.09$). As expected, PHA variation in chicks was only significantly explained by body mass (slope: $-0.06 \pm 0.02$; $F_{1,29} = 5.05$; $P = 0.03$), as PHA response was lower the heavier the chick.

Body mass change in chicks from vaccination to sampling date (4–11 days of age) was independent of mother's and chick's NDV-Ab titres (all $P > 0.30$). Likewise, neither treatment, sex nor the blood metabolite variables were related to body mass change (all $P > 0.2$). As expected, elapsed days between vaccination and sampling (slope: $1.20 \pm 0.28$; $F_{1,20} = 18.23$; $P < 0.01$) and tarsus length had a significant influence on body mass change (slope: $0.60 \pm 0.27$; $F_{1,63} = 4.96$; $P = 0.03$) (Table 1). House sparrow chicks with larger tarsi grew heavier independently of their mother's or their own level of NDV-Ab, once the period between measurements was accounted for.

Chick's survival from 4th day of age until fledging was unrelated to neither the maternal ($F_{1,22} = 0.76$; $P = 0.39$), nor the chick's treatments ($F_{1,23} = 0.31$; $P = 0.59$), and only body mass at the time of vaccination was related to fledging success (slope: $0.49 \pm 0.11$; $F_{1,89} = 18.29$; $P < 0.01$).

## DISCUSSION

House sparrow mothers developed a significant specific humoral response when challenged with NDV vaccine before egg-laying, and as a result their offspring were more likely to develop a specific humoral response when challenged with the same antigen. However, inter-individual variation in maternal NDV-Ab was the main determinant of chick specific immune response to NDV, as only chicks from mothers with high NDV-Ab concentration significantly increased their response to NDV vaccine (Fig. 1).

Maternal Ab transfer may reflect a dynamic balance between the benefits of providing an early specific protection and the costs of blocking the nestling's immune development (*Garnier et al., 2012*). In our study, maternal Ab could not be detected in 11-day-old chicks by means of standard diagnostic techniques, as no differences were found between control chicks from vaccinated and control mothers. The results suggest that maternal Ab transmission in house sparrows has a priming beneficial effect on the specific response of pre-fledging chicks (11 days old), by stimulating the endogenous production of Ab when early exposed to same pathogens as their mothers. This priming effect may be explained by a higher and/or faster antibody production in chicks exposed to maternal Abs. However, disentangling such non-mutually exclusive processes would require a detailed temporal monitoring of NDV-Ab response after fledging, which is beyond the focus of this study.

On the other hand, mothers influence offspring immunity in other ways than transmission of specific Ab, by providing other components through the yolk or by modulating parental investment. Therefore, it is possible that the enhanced specific immune response in offspring could result from other maternal effects than the transmission of specific Ab, or a combination of them. However, the fact that chicks did not experience any change in growth rate or PHA-response which is a complex inflammatory and immune response suggests that maternal effects were antigen-specific and not condition dependent (*Ahmed et al., 2007*).

Maternal treatment per-se was ineffective in explaining offspring specific response, and only when considering inter-individual variation in specific Ab levels such relationship was apparent. This result can be explained by several non exclusive reasons. First, the different response among breeding females to the experimental challenge, which led to a significant inter-individual variation in circulating Ab level, could arise from the usual individual variation to experimental vaccinations (*Zinkernagel, 2003*). Second, some breeding females could have been previously exposed to the virus in the wild (see *Broggi et al., 2013*). Previous exposure would lead to different maternal level of circulating Ab and in turn affect passive transmission of Ab to the offspring. In fact, one control chick originating from a control mother presented NDV-Ab, implying a natural exposure to the antigen (Fig. 1). Second, it is possible that laying females effectively transfer Ab only when their own systemic levels are above certain threshold (*Grindstaff, 2010*). Alternatively, it could be that maternally transmitted Ab do not persist long in the nestling blood, and after 11 days of age they are not detectable anymore (*Grindstaff, Brodie & Ketterson, 2003*; *Nemeth, Oesterle & Bowen, 2008*). The fact that breeding females were evaluated a few weeks after being challenged, while their chicks were challenged and sampled after a few days may rend our results

conservative as it may be that maternal effects are most effectively transmitted earlier after their challenge, or the effects appear after 11 days of chick age, but see *Garnier et al. (2012)*. The present results add to the accumulating evidence that exposure at different times during the course of a decay of maternal antibodies, which can be short-lived in altricial species like sparrows may affect chick immune response differently (*Zinkernagel, 2003*).

The few studies on maternally transmitted Ab on offspring specific immunity on wild avian species are not conclusive as the effects often vary among host species, pathogens and timing of exposure (*Zinkernagel, 2003*; *Hasselquist & Nilsson, 2012*). The limited knowledge on the persistence of maternal Ab, the temporal pattern of Ab production and the long-term consequences of maternal transmission limits our capacity to interpret these differences (*Boulinier & Staszewski, 2008*). Some studies report short-term enhancing effects (*Grindstaff et al., 2006*; *Pihlaja, Siitari & Alatalo, 2006*), while others have revealed a blocking effect (*Gasparini et al., 2009*; *Staszewski et al., 2007*; *Staszewski & Siitari, 2010*; *Garnier et al., 2012*; *Merrill & Grindstaff, 2014*). Interestingly, same host species can respond differently to maternally transmitted Ab when exposed to varying antigens (*Gasparini et al., 2006*; *Addison, Ricklefs & Klasing, 2010*). Other studies have found maternal transmission of Ab to buffer costs of an immune response (*Grindstaff, 2008*), whereas others found the effect of maternal Ab to be negligible (*Nemeth, Oesterle & Bowen, 2008*; *King, Owen & Schwabl, 2010*). Finally, long-term effects of maternal Ab-transmission on the specific immune response of chicks to antigens administrated on the previous breeding season to pre-laying mothers have been reported blocking (*Staszewski et al., 2007*), or enhancing the offspring specific response (*Reid et al., 2006*). In our study we found house sparrow mothers to enhance their offspring specific humoral response during the pre-fledging period, but not their PHA response, implying that specific Ab are transmitted without conditioning other components of their immune system like those involved in the response to the PHA test. These contrasting results highlight the varying effects that maternal Ab may have among hosts exposed to different pathogens, and the need to analyse differences in the dynamics of antibody circulation, timing of exposure to pathogens and the interaction with other maternally transferred components.

Interspecific comparisons suggest that larger and longer lived species, which also experience slower developmental times, may rely more strongly on the maternal effects that may be transmitted in larger amounts and persist for longer periods (*Garnier et al., 2012*; *Ramos et al., 2014*). In line with this suggestion and in contrast with our results, it has been argued that maternal transfer of Ab in fast-developing altricial species (e.g., house sparrow) may be limited or not even play any relevant role as endogenous Ab production is achieved soon after hatching, leaving maternal Ab a too short time-frame to be effective (*King, Owen & Schwabl, 2010*). Alternatively, maternal Ab may be effectively transmitted and still remain undetected, as they may persist for a too short time period or because of practical constraints as good samples are difficult to obtain from hatchlings of those small-sized species (*Lozano & Ydenberg, 2002*; *Nemeth, Oesterle & Bowen, 2008*). Furthermore, methodological differences among studies may partly explain the varying results on the effects of the maternal transmission of Ab on offspring specific immunity. Although most studies tested the effects by studying both mothers and offspring exposed to the same

antigen, this was not always the case (e.g., *Gasparini et al., 2006*; *Pihlaja, Siitari & Alatalo, 2006*; *King, Owen & Schwabl, 2010*). Our results suggest that maternal transfer of antibodies in the house sparrow is important enough to prime chick specific immunity, despite not being detectable by usual immunological techniques. The effect of maternally transmitted Ab appears to be host-pathogen specific, and dependent on the development of host humoral immunity at the time of infection. Furthermore, such effect could be of a hormetic nature (*Costantini, Metcalfe & Monaghan, 2010*), irrelevant under certain threshold and priming or blocking according to the different concentration of Ab transmitted and the individual condition (*Lemke, Coutinho & Lange, 2004*).

The ontogeny of the constitutive immunity is complex, influenced by both maternal and endogenous factors, with long term implications for the individual development and overall immunity (*Butler & McGraw, 2011*; *Van der Most et al., 2011*; *Arriero, Majewska & Martin, 2013*). Maternal transfer of specific Ab could benefit offspring by enhancing the specific humoral response while permitting growth rate to be maintained (*Grindstaff, 2008*), as found in this study. However, we found chicks developing stronger specific humoral response experienced decreased carotenoid levels and antioxidative capacity in blood. In addition to Ab, parent females transfer carotenoids and antioxidants to their eggs according to diverse endogenous and environmental factors such as mate attractiveness (*Saino et al., 2002*), own condition (*Hammouda et al., 2012*), pathogen abundance (*Gasparini et al., 2001*) or predation risk (*Morosinotto et al., 2013*). We found no differences in blood carotenoids or TAC measured in females either at challenge or post-challenge sampling times in relation to any of the treatments or NDV-Ab concentration, suggesting that variation in chick's levels arose from chick's own physiological adjustments rather than differential maternal transmission. Several studies have shown costly aspects of immunity in terms of antioxidative balance and carotenoid levels. Interestingly, other studies found carotenoid levels to be positively related to humoral response but unrelated to PHA response (*Bédécarrats & Leeson, 2006*). Carotenoids appear to compensate costly immune responses to pathogens (*Ewen et al., 2009*), or become depleted the stronger an immune response (*Saino et al., 2003*). In our study, house sparrow offspring fledged independently of their treatment or NDV-Ab concentration, suggesting that supposed advantages of enhanced specific humoral response, or costs of decreased antioxidative capacity and carotenoids in blood are to be experienced on a longer-term. However, the final cost-benefits balance derived from maternal transmission of antibodies will depend on the rate of exposure to pathogens, and the fitness costs derived from pathogen exposure in chicks with and without maternal antibodies.

In summary, we found that the specific humoral response to NDV in 11-day-old house sparrow chicks is enhanced by maternal exposure to NDV, most likely through passive transmission of specific Ab. However, nestlings investing in specific NDV-Ab production present lower carotenoid concentration and impaired antioxidative capacity in blood, suggesting that maternal priming of specific humoral response can be beneficial but may come to a physiological cost in pre-fledging small altricial species.

## ACKNOWLEDGEMENTS

We thank Marta Aliseda, Juan Luis Barroso, Oscar González, Cristina Pérez, Airam Rodríguez and Matthias Vögeli for their help during the fieldwork. We are indebted to Francisco Miranda and Olaya García for their work with sample analysis at the Ecophysiology laboratory in EBD. Plácido and Maribel from la Cañada de los Pájaros gently allowed us to work in their property. Albert Pagès and HIPRA personnel provided advice and technical support on NDV vaccine use and analysis. Jan-Åke Nilsson, Romain Garnier and one anonymous referee provided valuable comments on earlier versions of the manuscript.

### Funding

JB was funded by JAE/Doc and Juan de la Cierva postdoctoral grants by CSIC, the Spanish Ministry of Science and EU FEDER program. This work was partially supported by projects RNM118, RNM157, P07-RNM-02511 and P11-RNM-7038 of the Junta de Andalucía and the Spanish Ministry of Science project (CGL2009-11445 and CGL2012-30759). The funders had no role in study design, data collection and analysis, decision to publish, or preparation of the manuscript.

### Grant Disclosures

The following grant information was disclosed by the authors:
CSIC.
Spanish Ministry of Science.
EU FEDER program.
Junta de Andalucía: RNM118, RNM157, P07-RNM-02511, P11-RNM-7038.
Spanish Ministry of Science project: CGL2009-11445, CGL2012-30759.

### Competing Interests

The authors declare there are no competing interests.

### Author Contributions

- Juli Broggi conceived and designed the experiments, performed the experiments, analyzed the data, wrote the paper, prepared figures and/or tables, reviewed drafts of the paper.
- Ramon C. Soriguer and Jordi Figuerola conceived and designed the experiments, analyzed the data, contributed reagents/materials/analysis tools, reviewed drafts of the paper.

### Animal Ethics

The following information was supplied relating to ethical approvals (i.e., approving body and any reference numbers):

All procedures were approved by the EBD ethical committee (N/RefS.:G YB/AFR/CMM) and complied with current Spanish laws.

## Ethics

The following information was supplied relating to ethical approvals (i.e., approving body and any reference numbers):

All procedures were approved by the EBD ethical committee (N/RefS.:G YB/AFR/CMM) and complied with current Spanish laws.

## Field Study Permissions

The following information was supplied relating to field study approvals (i.e., approving body and any reference numbers):

Estación Biológica de Doñana, CSIC.

All procedures were approved by the EBD ethical committee (N/RefS.:G YB/AFR/CMM) and complied with current Spanish laws.

## Data Availability

The data is provided as Data S1.

## Supplemental Information

Supplemental information for this article can be found online at http://dx.doi.org/10.7717/peerj.1766#supplemental-information.

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
