# Peer review of "Transgenerational effects enhance specific immune response in a wild passerine"

_PeerJ, doi:10.7717/peerj.1766_

## Round 0.1 · original submission · Major Revisions

The two referees liked the manuscript and found the topic interesting and timely. However, they were also concerned that the writing should be improved. Both referees also felt that a major weakness of the study was that the “timing of events” has not fully been taken into account to design the work and to interpret the results. For instance, how relevant is it to sample 6 days post-vaccination instead of, let’s say, 10 days? How the results might change if chicks were vaccinated at a different age and so on. I do understand that the experimental design also depend on the specific constraints linked to the biological system studied (it is not possible to sample chicks that have already fledged), but I agree with the referees that the importance of “timing of events” should at least be better discussed. The referees also had a long list of comments and suggestions as to improve the manuscript that should be fully addressed in the revised version.

·

Basic reporting

I think the writing needs some attention to make the manuscript clearer. In many instances, the text could be shortened which would give more weight to the main ideas being expressed. In contrast some details are sometimes missing. For instance, the details of the number of females and chicks for each treatment group would be generally helpful. I think it would also be interesting if the authors could develop the role of the timing of exposure to the vaccine (or to a parasite in natural conditions) in the role of maternal antibodies.

Experimental design

I believe the methods need a lot of clarification. For instance, I understand that the chicks were vaccinated at 4 days old (l. 148-150) and then blood-sampled at 10 days old (l. 153-155). However, later it is mentioned that there was variability in the time interval between treatment and sampling. The range given (5-12 days, l.212) means that chicks were not sampled at 10 days old but anywhere between 9 and 16 days old, with an average around 11-12 days. This variability should be made clear throughout the methods, and its effects should be at least acknowledged in the results. For instance, one can wonder whether the age of the chicks has any effect on the immune response (i.e. do the chicks that had a longer time to mount an immune response have higher anti-NDV levels?).
The assignment of chicks to either treatment or control represents another potential methodological issue as it is currently described l. 150-151. It indeed sounds like chicks were not randomly assigned to the treatment group but rather based on their body mass. This may not influence the results much, but the process should be better described. It currently does not allow a proper understanding of the methods.

Validity of the findings

I believe the study to be an interesting one, and the data to support the conclusion of the possible existence of a priming effect of maternal antibodies in this species.
However, since the effect not very well supported, the authors may want to be cautious in their interpretation and the discussion of this result. I provide some suggestions in my general comments to achieve this (see the general comments).

Additional comments

Below are some more detailed comments that would help improve the manuscript:
Line 51: “important” rather than “relevant”?
Line 59-60: The humoral immune response does not take long to be effective upon first encounter with an antigen although it is indeed faster upon re-exposure to the same antigen. For instance, in chickens, it takes about 5 days to start detecting IgY after exposure to E. coli (Iseri and Klasing, 2013, Developmental & Comparative Immunology).
Line 80: “innate (inflammatory)” and “acquired (humoral)” need to be changed. The innate immune response cannot be reduced to the inflammatory response only, and the same is true of the acquired immune response (which does not include just the humoral response).
Line 83-84: The studies cited here seem to show some effects related to the immune system, not “little evidence”. Rephrase.
Line 87: “the intensity of THE immune system”
Line 112: “viral antigen” and not “viral infection” – the NDV vaccine is not a live virus.
Line 116: PHA is hardly a marker of “general immunity”. Rephrase.
Line 117-118: From the rest of the introduction, it sounded like the roles of carotenoids and antioxidants was to be tested, not merely controlled for. Maybe rephrase.
Line 124:125: Provide some examples of such pathogens.
Line 163: I am not sure I understand the idea of increasing the variance in female vaccination response. Would including the data obtained from the first two breeding attempts modify the results showed here?
Line 166: This would be a good place to give details on the number of chicks for treatment.
Line 171: “clone”
Line 179: at “10 days of age” (similar changes should be made throughout the manuscript)
Line 182-185: This is very confusing. Line 183 implies that the wing web width was read 24hrs after injection, but seems to be contradicted in the following line which states that the measures were made 25-45hrs after challenge. This should be clarified.
Line 187-188: All three measures were taken at the same time?
Line 201-202: Total proteins is not just a measure of nutritional condition, it may also reflect an important part of antioxidant activity. Albumin, a big component of total proteins, has important antioxidant properties (see for instance Roche et al., 2008, FEBS Letters).
Line 255: Does this result include chicks from unvaccinated mothers? There would be no reason to expect a correlation in these chicks.
Line 264-266: This effect is close to significance, but still non significant. In addition, it appears that the effect size is not large either. I would thus encourage the authors to remain cautious in their discussion of this effect. For instance, nowhere do the authors talk about potential issues with the vaccination process/efficacy.
Line 287-292: These results are pretty straightforward (growing chicks are getting bigger and larger, and the bigger are also larger). Not sure it’s worth mentioning the weight and tarsus length besides the fact that they are not influenced by the treatment of the chick (which would be an interesting result).
Line 308-310: This seems like a bit of an overinterpretation. The interaction is not significant and has a limited effect size. Although it may indeed indicate the potential for a priming effect, the underlying mechanism remains unclear (and the current manuscript does not offer any insight on that). In addition, there is no reason to expect direct transmission of antibodies to occur that late after hatching to directly increase the immune response of the chicks.
Line 312: “or” and not “of”
Line 321-325: The variation in the antibody response does not necessarily reflect previous level of antibodies. However, it would indeed be useful to know about the circulation of NDV in the population during the course of the experiment. For instance, the fact that some unvaccinated chicks had antibodies indeed indicates that the virus is circulating. But is there any brood effect on the infection status (i.e. do vaccinated chicks in the same brood also display higher levels of antibodies)?
Line 325-326: All evidence seems to indicate that the transfer of antibody to the yolk in females is passive. Low levels in females would result in low levels in chicks, which may thus become rapidly undetectable.
Line 328-332: I am not sure I understand the argument here.
Line 335-336: This seems very general, and there are quite a few studies (reviews cited in the manuscript – Boulinier & Staszewski 2008, or Hasselquist & Nilsson 2009) that seem to disagree with that statement.
Line 337-338: Remove “unspecific” in front of “humoral response”.
Line 341: Remove “synthetic”
Line 344: This effect has also been reported in Cory’s shearwater and NDV (Garnier et al. 2011, cited in the manuscript).
Line 345-347: Rephrase. I guess the authors are trying to say that in some species, exposure of chicks with the same parasite may result in either a blocking effect or a priming effect. This highlights the need to discuss effects of the timing of exposure to the antigen.
Line 352-353: Staszewski et al. (2007) report a blocking effect, not an “impaired immune response”.
Line 394-395: The variation could also arise from differences due to parental provisioning during growth.
Line 412: I am not entirely which developmental cost the authors refer to as none of the chicks, irrespective of their treatment or immune response, showed any differences.

Reviewer 2 ·

Basic reporting

The article is overall written in good scientific English, but the Introduction and Results would benefit to be rewritten to outline and justifiy more clearly how the work fits into the field of knowledge and how the results fits with what was expected given the study design. See General comments.

Experimental design

The knowledge gap being investigated should be better identified, and clearer statements should be made as to how the study contributes to filling that gap. See General comments.

Validity of the findings

The conclusions presented in the current version of the manuscrit are too confidently stated given the presented results and the way the study design is justified. See General comments.

Additional comments

This manuscript reports the results of an experimental study done with wild sparrows in which measurement of antibody levels were conducted following vaccination of females and their offspring against a specific antigen (NDV). The design of the experimental approach is not novel, but the topic is important because very little is still known about the dynamics of maternal antibody transfer and persistence in passerines, and the potential effects of maternal antibodies. A major weakness of the study is nevertheless that the timing of events does not seem to have been considered carefully by the authors, despite its obvious expected importance. It is not only important that maternal antibody might have potential ‘blocking’ or ‘priming’ effects (Hassequist & Nilsson 2009), but also how this could happen, and it is important to realize that timing and mechanistic issues are involved and need to be considered/justified when designing studies on such a topic (Boulinier & Staszewski 2008 - cited). The results are also based on limited sample sizes given the number of variable considered and the apparent natural exposure to NDV in the study population is lowering the strength of the inference that can be made from the experimental approach.
I outline below three questions which should have been considered more clearly with regards to the design and expected results:
(1) How long was it expected for a sparrow chick to mount a quantifiable specific humoral immune response following exposure to the vaccine? Is 6 days a logical guess? This point is not discussed despite its critical importance given the fact that blood sampling for the quantification of antibodies was done only 6 days post-vaccination in chicks, and not after. Wouldn’t 15 days post-vaccination have been a better timing? What is the justification (apart a practical one) that the sampling was done only 6 days post vaccine infection? Does it limit some of the inference that can be made?
(2) Also, how long after hatching was a quantifiable level of specific maternal antibody expected to last in sparrow chicks and could this matter for testing a potential blocking versus priming effect? This requires to be considered seriously early in the manuscript to justify the study design, potential predictions and interpretations (the study design involved vaccination of 4-days old chicks from vaccinated/non-vaccinated mothers).
(3) Finally, how fast were females expected to respond to the vaccine and could this have affected the amount of antibodies transferred to the different chicks of each brood?
As in several ‘behavioural/immunoecology’ papers involving maternal antibody transfer in small passerines, no or little specific maternal antibody was reported to be detected in the blood of the sampled chicks (Lozano & Ydenberg 2002, Nemeth et al. 2008 - cited, King et al. 2010 - cited), which is likely due to a combination of two things: (i) a possible very short persistence of maternal antibodies in those species and (ii) the small size of young hatchlings of those species, precluding easy measures of plasma content shortly after hatching. Shouldn’t this be said clearly somewhere in the manuscript ?
Other comments:
Abstract: There is a need to specify more clearly some of what was done and why a vaccine was used. At what age was it possible to test whether vaccinated chicks from vaccinated mothers developed a strong NDV antibody response? Maternal antibody level was measured when?
Line 39: Delete ‘s’ at the end of ‘other immunity components’ because only a measure of PHA was used in addition to the antibodies specific to the vaccine (this goes also for later on in the manuscript, e.g., line 358). If the authors believe that by using a PHA test they test several components of immunity, then they should outline how a negative result can be interpreted as a lack of effect on other components of immunity.
Line 50: Why not citing two earlier reviews there ? (Grindstaff et al. 2003 & Boulinier & Staszewski 2008)
Line 71: ‘remains controversial’ ? What does this mean? How? A little more should be said here, notably regarding the potential importance of the timing of events (see lines 148-150).
Lines 114-115: ‘aspects’ plural and ‘response’ singular ? One or several other aspects of immunity was explored? (see above)
Line 135: how many days between broods? (notably between the first and second, given the study design). How fast are females expected to respond to vaccination? It may be interesting to have a look at the supplemental material of Midamegbe et al. 2013 for some data on egg yolk NDV antibody levels following vaccination of females at different times before laying.
Lines 148-149: What is the aim of the experimental test with a vaccination at 4 days of age? (testing a potential priming or blocking effect? Or both, but then how, given what might be expected of the decaying of maternal antibodies?)
Line 212: 7 days is relatively short for expecting a detectable humoral immune response, no?
Lines 221-223: Awkward. Needs rewording. Also, a lot of covariables are considered: is this approach removing potential problems associated with considering a large number of variables relative to the sample size?
Line 246: Please report estimates (+- SE) rather than only F and P values.
Line 251: Were any of these results predicted/expected? Shouldn’t this be said (‘As expected…’) to help the reader? The remark goes for the rest of the Results.
Line 298-300: Too affirmative given the results that are presented and the study design.
Lines 306, 332, 365: ‘2012’ not ‘2011’ for Garnier et al.
Line 312: ‘or’ not ‘of’.
Lines 323-325: This messes up the experimental test, no?
Line 335-336: shouldn’t this statement be limited to wild passerine species ?
Line 344: Hasn’t this been tested in another seabird species, the Cory’s shearwater (and via an exposure of chicks to the vaccine at 20 days post hatching)? See Garnier et al. 2012. A discussion about the timing of the test of a blocking/priming effect is required, no?
Line 353: Was it really an ‘impaired’ specific immune response to NDV vaccine or more a ‘blocking’ effect, showing that the maternal antibodies were functional. ‘Impaired’ seems to suggest that this study showed that maternal antibodies did not have a functional role.
Lines 361-362: ‘substantial variation among species’ with no logic? Where are we going from there?
Lines 376-380: What about studies/reviews discussing timing issues with regards to blocking/priming effects (e.g., Lemke et al. 2004).
References cited above but not in the manuscript:
Lemke et al. 2004. Lamarckian inheritance by somatically acquired maternal IgG phenotypes. Trends in Immunology 25: 180-186.
Lozano, G.A. & Ydenberg, R.C. 2002. Transgenerational effects of maternal immune challenge in tree swallows (Tachycineta bicolor). Canadian journal of zoology 80: 918-925.
Midamegbe et al. 2013. Female blue tits with brighter yellow chests transfer more carotenoids to their eggs after an immune challenge. Oecologia 173: 387-397.

---

## Round 0.2 · Minor Revisions

Both referees found this version of the manuscript to be much improved compared to the previous one. They however still had a long list of comments and suggestions as to further improve the clarity of the manuscript. These comments and suggestions should be easily addressed in a second revision.

·

Basic reporting

(Note that the line numbers used in this review are taken from the version of the manuscript including track changes.)
I appreciate the efforts of the authors to clarify the manuscript, but I still feel that the manuscript could be shortened somewhat and clarified some more. I provide specific comments below to that end, but generally speaking the introduction and the discussion suffer from numerous repetitions (see for instance l.123-150: the second paragraph has a significant amount of overlap with the first one). Following my comment on the first version of the manuscript, I still believe that providing the details of sample sizes in the text could be very helpful, especially because I find it relatively uneasy to readily access this information in Table 1.
It also appears that no mention is made of the raw data availability, despite it being provided as a supplementary file. A mention of this availability could be made in the Material and Methods section to fully comply with PeerJ policy.

Experimental design

I do agree with reviewer 2 that the knowledge gap was not particularly well defined in the first version of the manuscript. The changes made on the introduction have improved this, but I still feel that the specific question investigated could be better stated, maybe in the last paragraph of the introduction. Adding predictions could be one way to achieve this.

Validity of the findings

I have no concerns with the validity of the findings. However, I think that some aspects need to be better discussed. This particularly includes the timing of the exposure to the parasite, and how it influences the type of effect of maternal antibodies. I would for instance recommend the use of references such as Zinkernagel (2003, Annu Rev Immunol, 515-546) to help discuss how exposure at different times during the course of the decay of maternal antibodies may have different effects on the immune response of the chick.

Additional comments

From here on, the comments are line by line, with line numbers taken from the manuscript with track changes.
l.9: Running title could be improved. Maybe something along the lines of “Maternal effects on passerine immunity”?
l.15: Maybe this would be a better place to include the permit number currently l.339-340?
l.39: (here and elsewhere throughout the manuscript). “10-day-old chicks” (with hyphens)
l.41-42: Not really 5 weeks, as there is a range. This could be replaced by something like “measured during chick-rearing”.
l.44: “in” instead of “on”
l.44-46: I don’t really follow the logic in that sentence. How is not detecting maternal antibodies suggestive of a priming effect? Rephrase.
l.47-49: This sentence is only generalities about maternal antibodies, none of which are the focus of this manuscript. It should be replaced by a conclusion about the current study.
l.59-60: Rephrase or delete.
l.65: “These” instead of “such”.
l.72: “response” instead of “reaction”
l.73: “a” few days
l.80-81: As mentioned in the sentence, maternal antibodies can be detected up to several weeks after hatching in some species (Cory’s shearwater in Garnier et al. 2012, but see also Chang et al. 2007 Vaccine in vultures). This hardly corresponds to the definition of “generally short-lived”. Please rephrase.
l.81 (and elsewhere in the manuscript): “Garnier et al. (2012)” is the correct reference, as correctly pointed out by reviewer 2. The article was published online in Dec 2011, but is in the May 2012 issue (vol 279, p. 2033-2041).
l.92: “the initial amount transferred” – remove “of” and add “initial”. The importance of the initial amount for the persistence is what Grindstaff 2010 is all about.
l.92: Remove “the effects of”
l.96: I am not sure “controversial” is the correct word here. Maternal antibodies have been shown to potentially have a blocking or a priming effect depending on the timing of exposure, but these mechanisms are not mutually exclusive. This is thus hardly a “controversy”. This was already pointed out by reviewer 2 in his/her initial review.
l.97: “a” few days
l.98-100: This is unnecessarily repeating l.73-76 (with the same supporting reference).
l.100-101: The experimental evidence is neither “meagre” nor “scarce”. There is actually a somewhat important body of work on the transfer of maternal antibodies in avian species, with lots of references to be found in the review articles/book chapter cited in the manuscript.
l. 106: Not sure this is the best reference available on the potential costs of immunity. General references. Viney et al. (2005, Trends Ecol Evol) or Lochmiller & Deerenberg (2000, Oikos) would seem more appropriate for such a general statement.
l.113: Horak et al. 2006 – this reference does not appear in the list of references. Presumably J. Exp. Biol 2006?
l.113-115: I appreciate the effort to accommodate my previous comment, but I still don’t understand that sentence.
l.123: “the” instead of “such”
l.133-137: this reads very similarly to l.123-125.
l.141-146: This could be shortened and clarified. Mothers transmit carotenoids and antioxidants in the egg yolk and provide these molecules through diet after hatching, and both are likely to stimulate the chicks own immune response.
l.146-148: Seems redundant with l.126-128
l.148-150: Reads as l.123-125, with the exact same reference.
l.157-158: There is no measure of “mother condition” in this study. Do the authors mean “vaccination status” here?
l.172: “Newcastle Disease Virus”
l.174: that causes “a” highly contagious disease
l.181: “year”
l.182: Separate in two sentences for clarity, after “asynchronously”.
l.198-199: I am not sure what the pilot study has to do with this protocol.
l.204: This is a quite important addition as females having received a booster vaccination will have higher antibody levels and thus will be transmitting more antibodies.
l.211: “pers. obs.”. In addition, whose personal observation is this?
l.211-212: This is another quite important clarification. If I understand correctly the experimental design all chicks were vaccinated at 4 days old. Then, when the mean brood age was 10 days old, all chicks were blood sampled to measure anti-NDV antibodies. This means that some chicks would have been 7-8 days old, and would only have had 3-4 days to mount an immune response. Given that it takes 5 days to start detecting IgY in adult birds, this makes it unlikely that these chicks would have had enough time to mount a response, irrespective of the presence/absence of maternal antibodies. In vaccinated chicks only, is the probability to detect an antibody response dependent on the age of the chicks (or alternatively on the time between vaccination and measure)?
l.238: What is the amount of sera used to perform the serial dilution? This information is not provided in Broggi et al. 2013 either.
l.248: I suspect this also means that the mean brood age would have been 10 days old?
l.270-271: What are the adaptations? If described elsewhere, please provide reference.
l.291: “7.72 +/- 0.19 days” – this means that on average broods were 11.72 days old? How does that relate to the 10 days average brood age mentioned l.212? Same question applies to the 24-65 days range, which does not match the 2-7 weeks mentioned l.231 (this range implies 3-10 weeks). Please modify so that these numbers match.
l.332: Reference table 1 here for details of the numbers for each laboratory test performed.
l. 335-336: Do the authors mean antibody titers here? These may need log transformation (although there should be no zeros, so not log +1). I am not sure why other blood metabolites would need log transformation to achieve reasonable normality.
l.346: “or” instead of “nor”
l.347: begin sentence with “as expected”, “increased” instead of “raised”
l.355: “More surprisingly” (or other similar) instead of “likewise, and against the expectations”.
l.360-361: What is an interaction of days between mother and chick sampling supposed to represent? What would be the expectation for such an interaction?
l.365: “associated with” instead of “related to”
l.370: In addition to the previous covariates (which remain significant) or do the previous covariates lose significance?
l.386-388: What is used as a PHA response – ‘raw’ thickness of the wing web or a % change of the wing web thickness? The prediction for an effect of size/mass would probably be different depending on which measure is used.
l. 396: “As expected” at the beginning of the sentence.
l.414: “in” instead of “on”
l.416: “detected” instead of “appreciated”
l.416-419: I don’t understand that sentence. It is simply untrue that chicks from control mothers could not mount an immune response. (1) Antibodies could be detected in some of them. (2) The fact that antibodies could not be detected at the time of sampling, which was still pretty close to the time of vaccination, does not mean antibodies would not have been detected later. As both I and reviewer 2 mentioned in our respective reviews, there is a need for caution when discussing these results.
l.427: through “the” yolk
l.441: This description of the PHA response is inaccurate: it does not measure only an immune response, and the immune component of the response is not just innate. Replace with something along the lines of “a complex measure of inflammatory and immune responses”.
l.446-450: I am not sure this is the most obvious explanation. Variations in responses following vaccination is commonly reported, and the range visible in figure 1 does not seem different from expected variations. The absence of response in 1/several individuals also begs the question of whether this could be due to manipulation errors, and whether these may be more likely to happen in chicks.
l.451: The correlation between maternal NDV levels and the levels of these antibodies in chicks is well acknowledged in the literature – see Garnier et al. 2012 or Ramos et al. 2014.
l.454-456: The persistence of maternal antibodies in sparrows has been shown to be probably very short with a half-life of 3 days and no antibodies against West Nile Virus detected after 9 days (Nemeth et al. 2008 – cited by reviewer 2).
l.456-460: Mothers transmit IgY which persist in blood for much longer than a few weeks after challenge. For instance, a single NDV vaccination in pigeons is expected to be protective for at least a year. In the wild, Cory’s shearwater females maintain detectable IgY levels for years following a single vaccination (Ramos et al. 2014, Am Nat). The authors should discuss this timing in relation to, for instance, the dynamics reported in Midamegbe et al. (2013).
l.473-532: This whole paragraph could be shortened a lot, and clarified. The species list is pretty long and there are details that are probably not really necessary. In addition, this paragraph would be a good place to also discuss the effects of the timing of exposure. For instance, the fact that some species can display either a blocking or a priming effect should be discussed in this light (l.487-490).
l.524: Here again, the timing of exposure of kittiwakes to NDV is probably crucial for the blocking effect. It is well discussed in the reference by Staszewski et al.
l.586-600: This is a very affirmative sentence given the evidence presented in the manuscript. TAC and carotenoids were not measured in females at the time of egg laying and in chicks just after hatching, precluding this sort of definitive statement. Again, there is a need for caution in the way the results are discussed.

Reviewer 2 ·

Basic reporting

No comments

Experimental design

No comments

Validity of the findings

No comments.

Additional comments

The manuscript is reporting interesting results of a soundly designed study. This revised version of the manuscript is much improved compared to the original one.

I only have a few minor editorial suggestions, listed below with line numbers as on the track change version provided by the authors:
Line 69 : insert ‘a’ to make ‘in a few days’.
Line 147: change to make ‘, e.g., Newcastle Disease Virus’
Line 149: insert ‘a’ before ‘contagious disease’.
Line 181: comma after ‘hatching asynchrony’.
Line 193: insert ‘the” before ‘last’.
Line 378: insert ‘a’ before ‘complex’.
Line 384: insert ‘. This’ in place of the comma after ‘apparent’.
Line 390: insert ‘a’ before ‘control’.
Line 395: insert ‘a’ before ‘few’. Seemingly at the next line.

---

## Round 0.3 · accepted · Accept

All the concerns raised during the last round of review have been satisfactorely addressed in this revised version.